# A Study of the Impact of Digital Competence and Organizational Agility on Green Innovation Performance of Manufacturing Firms—The Moderating Effect Based on Knowledge Inertia

**Zhucui Jing [1], Ying Zheng [2,*] and Hongli Guo [2]**

[1] School of Economics and Management, Research Center for Central and Eastern Europe, Beijing Jiaotong University, Beijing 100044, China; zcjing@bjtu.edu.cn

[2] School of Economics and Management, Beijing Jiaotong University, Beijing 100044, China; 22120709@bjtu.edu.cn

* Correspondence: 23120735@bjtu.edu.cn

**Abstract:** Hierarchical regression is used to empirically investigate the impact of digital capabilities on green innovation performance, as well as the mediating role of organizational agility and the moderating effect of knowledge inertia. Based on the data from a large sample of 383 middle and senior managers from manufacturing companies, the dynamic capability theory is applied to SPSS 27.0. The results show that digital capability contributes to green innovation performance; knowledge inertia moderates the inverted U-shape between digital capability and green innovation performance; and two dimensions of organizational agility, market agility and operational adjustment agility, partially mediate the relationship between digital capability and green innovation performance. This paper contributes new ideas for companies to develop organizational agility, control knowledge inertia, enhance green innovation performance, and finally, sustainably gain a competitive advantage position.

**Keywords:** digital capability; knowledge inertia; green innovation; organizational agility

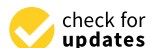



## 1. Introduction

Developing the economy and protecting the environment go hand in hand in order to escort the realization of high-quality and sustainable development of China's economy. China attaches great importance to the green development strategy; in July 2023, General Secretary Xi Jinping proposed at the National Conference on Ecological Environmental Protection building a green and intelligent digital ecological civilization and accelerating the promotion of modernization in the direction of harmonious coexistence between human beings and nature. Green innovation is an essential way for countries to achieve high-quality economic development (Lv et al. 2023). Enterprises focusing on improving green innovation performance could improve resource utilization and create greater ecological value while realizing economic value (Ogiemwonyi et al. 2023). The innovation of the green economy is closely related to information technology (Cao et al. 2023), and the government supports social actors to increase the use of digital green technology and promote the synergistic development of digitization and greening (Yu et al. 2022). Since 2020, the "Guide for the Digital Transformation of Small and Medium-sized Enterprises", "Digital China Construction Overall Layout Planning", and other policies have been introduced, and digitization has become the "infrastructure" of enterprise innovation and development. Therefore, how digitization capability can promote green innovation is an important issue to be solved in the new era.

With the development of the digital economy, how to build an "agile enterprise" has become a hot topic of discussion among scholars. Even with advanced science and

technology, enterprises must strengthen their ability to predict trends, collaborate with resources, and analyze strategies in order to remain competitive in the market (Yi and Cao 2022). Many scholars have focused on the role of organizational agility in promoting the competitive advantage of enterprises and tried to understand the path of its impact on performance. Jing et al. (2010) proved that organizational agility enhances sensitivity and responsiveness to changes by improving the business processes of enterprises, and thus improves the performance level of the enterprises. Tallon and Pinsonneault (2011) found that organizational agility has a significant role in the development of information technology and business processes. They found that organizational agility plays a mediating role between information technology and firm performance. However, the existing studies have not yet conducted an in-depth analysis of organizational agility as a mediating variable between digital capabilities and green innovation performance.

Innovation is the stage of knowledge creation (Nonaka 1994). In order to enhance green innovation capacity, enterprises should not only focus on knowledge management, consolidate the knowledge base, and strengthen the scientific research capacity but also apply knowledge more scientifically and efficiently. Chinese scholars believe that knowledge inertia is always realized via knowledge activities, and the behavior of knowledge activities is an important way to enhance the innovation capacity of enterprises (Li and Zeng 2019); however, Haroon and Shariff (2016) believe that knowledge inertia could have a negative impact on enterprise innovation. Apparently, there is still no academic consensus on the relationship between knowledge inertia and corporate innovation. Therefore, this paper has a certain research value to explore the moderating path of knowledge inertia in the relationship between digital competence and green innovation performance.

Based on the literature compilation, it is found that the established studies mainly explore the motivation of green innovation in terms of media opinion (Zhao and Zhang 2020), executive awareness (Cui et al. 2021), government subsidies (Zhang et al. 2023), and upstream and downstream of the supply chain (Zhang et al. 2021). In terms of the path of action, digital capability mainly empowers cost reduction and efficiency (Qi and Liu 2023), eases financing pressure (Zhang et al. 2023), improves information disparity (Zhang et al. 2021; Li et al. 2020), strengthens internal governance (Qi and Liu 2023), and attaches importance to environmental regulation (Cui et al. 2021; Yu et al. 2022) to stimulate green innovation in firms. However, existing studies are lacking in the role analysis of digital capabilities in combination with organizational agility and knowledge inertia. In addition, the research on the role of these three in the green innovation performance of enterprises needs to be further supplemented.

The research objectives of this paper contain the following four main points: (1) explore the relationship between digital capability and green innovation performance of enterprises so as to improve the empirical research on the factors affecting green innovation performance in China's manufacturing industry under the background of the digital economy; (2) confirm the role of organizational agility in the path mechanism between digital capability and enterprise green innovation performance, and innovatively propose the influence path of "digital capability—organizational agility—green innovation performance" so as to further uncover the "mechanism black box" of digital capability and enterprise green innovation in China's manufacturing industry; (3) construct a scenario analysis framework based on knowledge resources to discuss the impact of digital capability of manufacturing enterprises on their green innovation performance under different degrees of knowledge inertia so as to make the study on the impact mechanism of digital capability on green innovation performance more comprehensive; And (4) based on the existing research results, this paper provides reasonable countermeasures and suggestions for the Chinese government and manufacturing enterprises to enhance the performance of green innovation and realize the parallel improvement in economic value and ecological value of enterprises and also provides certain references for manufacturing enterprises in other developing countries to play the synergistic role of digital development and green transformation.

To sum up, based on the dynamic capability theory, this paper investigates the impact of digital capability on green innovation performance, focusing on the mediating role of organizational agility and the moderating role of knowledge inertia, taking into account China's local context and corporate culture.

## 2. Theoretical Basis

The concept of dynamic capabilities was introduced by Teece and Pisano (1994), who viewed dynamic capabilities as the internal and external capabilities that firms build, integrate, and reconfigure in order to adapt quickly to changing environments. Dynamics refers to the need for firms to continuously update their capabilities to adapt to the changing external environment (Teece et al. 1997); capabilities refer to the key role of strategic management in integrating and transforming firms' internal and external resources and organizational capabilities (Teece et al. 1997), and the theory reflects the skills that firms need to gain a long-term competitive advantage in a particular path or market position.

Existing research suggests that dynamic capabilities are closely related to knowledge. Subramaniam and Youndt (2005) suggested that knowledge resources are fundamental conditions that influence the formation and development of dynamic capabilities. Prieto and Easterby-Smith (2006) suggested that dynamic capabilities are formed and act on knowledge flows and knowledge management processes, which have a guiding effect on the enterprise's knowledge output and continuous update. The existing knowledge and experience of an organization are less likely to be directly valuable in a rapidly changing environment and need to be further transformed via dynamic capabilities as an enabler to make them effective (Zhao et al. 2021). Therefore, dynamic capability can be used as a theoretical basis to explore the role of knowledge inertia in the process of enterprise development.

The rapid development of digital technology has intensified the changes in the enterprise innovation environment, and in the increasingly changing digital environment, the dynamic capability theory perspective can provide a deeper understanding and exploration of the process of enterprise survival and development (Zhuang et al. 2020). Vial (2019) explored the enterprise digital transformation mode based on the dynamic capability theory, arguing that digital capabilities help enterprises eliminate the original outdated inertia thinking and behavioral dependence and optimize the realization of value creation. Khin and Ho (2018) argued that digital capability is essentially a dynamic capability of an enterprise, which is the ability to apply digital technology and management expertise in the process of developing new digital products. According to the theory of dynamic capabilities, the innovation activities of enterprises are affected by the dynamic capabilities of enterprises, and specific dynamic capabilities can promote the innovation activities of enterprises well (Wang and Ahmed 2004). Especially in the digital era, enterprises pay more attention to the dynamic enhancement of core competencies so as to better respond to environmental changes (Zhu et al. 2020). In summary, the dynamic capability theory can fully explain the connotation and formation of digital capabilities and form an important theoretical support basis for this paper to explore the role of digital capabilities on innovation performance.

## 3. Conceptual Background and Hypotheses

### 3.1. Digital Capability (DC) and Enterprise Green Innovation Performance (GIP)

With the development of the digital economy, more studies have recognized the conclusion that the digital capacity of enterprises could have a positive impact on green innovation performance: based on the national strategy dimension, enterprises' steps towards the digital economy are in line with the major strategy of "Digital China" implemented by the state, and they are more likely to enjoy the government's subsidies and grants when they start their business (Zhang et al. 2023). The reduction in financing pressure makes enterprises brave enough to assume social responsibility and strengthen the willingness for green innovation and development (Jiang and Lu 2023). Based on the Eco-environmental dimension, digital transformation can achieve green development and

environmental protection goals by cutting energy waste and consumption, and its nature is already embedded in co-development goals and drives green innovation performance (Yu et al. 2022). Based on the supply chain dimension, Zhang et al. (2021) found in an empirical study that supply chain digitization can improve the level of green innovation of enterprises, and then realize the improvement in environmental value and economic performance. Based on the enterprise type dimension, scholars found that digitization level and green innovation all showed a positive correlation by successively examining manufacturing industries (Yue et al. 2020; Wei and Sun 2021), resource-based enterprises (Wang et al. 2022), and heavily polluting enterprises (Song et al. 2022). Based on the technology and resource dimensions, the introduction of digital infrastructure and talent can motivate firms by reducing the risk of failure and cost of the green innovation process (Qi and Liu 2023), thus improving green innovation performance. In the process of product development, processing, and sales, the use of digital technology drives it to cultivate the concept of green R&D and reduce the cost of process flow and management services (Qi and Liu 2023). In summary, Hypothesis H1 is proposed as follows:

**H1.** *Digital capability promotes green innovation performance of enterprises.*

### 3.2. Digital Capability (DC) and Organizational Agility (OA)

Organizations need to have the ability to continuously monitor and respond to uncertain external environments. In the information age, market dynamics and trend forecasts are hidden in massive data resources. Enterprises need to identify, process, and integrate various data information to tap into existing or potential market demand in order to provide products and services that satisfy customers. For example, Alibaba and Amazon record consumer behavioral data to analyze customer preferences and launch relevant products with the help of big data technology (Zhou et al. 2023). Organizational agility has been measured in the literature in terms of speed of market perception and sensitivity of operational adjustments. Yi and Cao (2022) pointed out that digital capabilities help companies integrate decision-supporting data elements into the production and operation process chain and effectively analyze data across the chain to achieve market agility. Business operations based on outdated and old-fashioned production models could hinder digitization and reduce resilience; on the contrary, business units can enhance resource integration and sharing by integrating elements of technological innovation, thus improving enterprise agility (Yi and Cao 2022). In terms of operational adjustment agility, the aggregated use of IT resources (Gao and Li 2017), cross-functional departmental data integration (Qi and Liu 2023), and stronger management capabilities (Cui et al. 2021) could promote the operational agility of the enterprise. According to scholars Akter et al. (2016), the use of digital technologies, such as the Internet of Things, facilitates access to various types of data during business operations, improves the transparency of operational processes, and the ability to co-create and share knowledge resources, which, in turn, promotes operational adjustment agility. In summary, the hypotheses are proposed as follows:

**H2.** *Digital capabilities (DC) promote organizational agility (OA);*

**H2a.** *Digital capabilities (DC) promote market agility (MA);*

**H2b.** *Digital capabilities (DC) promote operational adjustment agility (OAA).*

### 3.3. Organizational Agility (OA) and Enterprise Green Innovation Performance (GIP)

In a dynamically changing competitive market, enterprises can quickly grasp first-hand information such as newly introduced environmental policies and external green technology development trends (Qi and Liu 2023). This facilitates enterprises to further capture consumers' green perceptions and preferences, so as to accurately perceive green development opportunities in the market, reduce the risk of green innovation, and improve

the performance level (Tian et al. 2023). In addition, in enterprises that have easier access to data and information about competitors, suppliers, customers, and other stakeholders, the flow of information passes and operates smoothly between departments (Gao and Li 2017), and these enterprises are more likely to adjust the business operation mode of energy saving and emission reduction in a timely manner (Song et al. 2022), which in turn stimulates the subjective initiative of enterprises in green innovation. From the supply chain perspective, enterprises can sense and obtain the green raw material information provided by upstream suppliers and the environmental demand information of downstream consumers in a timely manner to enhance the targeting of green innovation (Zhang et al. 2021). At the same time, real-time business adjustments promote more effective collaborative management decisions and the development of resource-efficient enterprises (Zhen et al. 2023). For example, enterprises could combine internal and external advantages to realize the sharing and application of green resources and find suppliers that could provide more environmentally friendly and healthy raw materials (Qi and Liu 2023). These optimization measures not only provide customers with green products but also help them to improve their environmental performance. These optimization measures not only provide better services for customers and enhance users' satisfaction with the environmental image of the enterprise but also make the operation process more efficient and flexible and easy to control costs (Qi and Liu 2023), which is conducive to the enterprise to improve resource utilization and build a green production system. As such, the hypotheses are proposed as follows:

**H3.** *Organizational agility (OA) promotes enterprises' green innovation performance (GIP);*

**H3a.** *Market agility (MA) promotes enterprises' green innovation performance (GIP);*

**H3b.** *Operational adjustment agility (OAA) promotes enterprises' green innovation performance (GIP).*

*3.4. Mediating Role of Organizational Agility (OA)*

After analyzing the results of a survey of 241 corporate executives, Tallon and Pinsonneault (2011) found that the effective application of information technology can stimulate corporate agility (Li and Zeng 2019), which in turn enhances the level of innovative research and development of a company's green products or services (Qi and Liu 2023; Song et al. 2022). Enterprises with higher digital capabilities can quickly target and utilize the advantages and value brought by IT advances (Yi and Cao 2022), which facilitates the efficient formulation of solutions regarding environmental pollution control, energy consumption control, and R&D investment in environmentally friendly products, and thus enhances the competitive position of eco-environmental protection. Organizational agility is generally measured in two dimensions: market and operational alignment (Lu and Ramamurthy 2011). Higher market agility implies that firms are able to significantly improve their awareness of the dynamic environment and their flexibility to respond to changes (Urbinati et al. 2019). In fact, after successfully acquiring real-time market information with the help of data capabilities, companies can conduct in-depth data analysis based on consumer behavioral preferences to develop differentiated and personalized marketing programs (Mikalef et al. 2020), which creates greater opportunities for companies to achieve green innovation (Tian et al. 2023). On the other hand, operational adjustment agility focuses on an organization's ability to quickly take alternative actions or make necessary internal adjustments in response to supply disruptions and market and customer changes (Lu and Ramamurthy 2011), which in turn affects team performance (Jing et al. 2010). Firms with higher digital capabilities tend to have higher internal change agility by relying on cutting-edge technologies to improve green innovation performance by innovating clean and harmless (Zhang et al. 2023), energy-saving and emission-reducing new technological equipment (Qi and Liu 2023), optimizing internal green innovation business processes (Zhang et al. 2023; Qi and Liu 2023), and improving resource allocation upstream and down-

stream of the supply chain (Zhang et al. 2021; Song et al. 2022). Therefore, the following hypotheses are proposed:

**H4.** *Organizational agility (OA) mediates the relationship between digital competence (DC) and firms' green innovation performance (GIP);*

**H4a.** *Market agility (MA) mediates the relationship between digital competence (DC) and firms' green innovation performance (GIP);*

**H4b.** *Operational adjustment agility (OAA) mediates the relationship between digital competence (DC) and firms' green innovation performance (GIP).*

*3.5. The Moderating Role of Knowledge Inertia (KI)*

Knowledge inertia is the attribute that organizations or individuals are accustomed to solving problems based on commonly used knowledge sources, customary methods, and existing experiences (Li and Zeng 2019; Liao et al. 2008). Liao et al. (2008) pointed out that knowledge inertia makes enterprises fall into the core competence dilemma and produce a certain degree of resistance to learning and applying new knowledge and divided it into two dimensions to construct a scale. Learning inertia emphasizes the process in which employees are driven by the willingness to learn and acquire new knowledge from the outside world to solve real-world problems (Liao et al. 2008; Cao et al. 2022); and empirical inertia refers to the fact that employees deal with problems according to existing internal procedural patterns and customary regulations (Liao et al. 2008; Li and Zeng 2019). When employees' learning inertia changes from weak to strong, their willingness to explore, acquire, and integrate cutting-edge digital information becomes stronger (Li and Zeng 2019). This characteristic helps the entity break the inherent knowledge cage based on operational processes (Yi and Cao 2022), thus strengthening the positive effect of digital capabilities on innovation performance (Akter et al. 2016). Driven by the willingness to learn, the enterprise focuses on expanding the digital knowledge source to obtain the corresponding external information (Cao et al. 2022) and at the same time, effectively improves the external environmental information perception and internal knowledge conversion efficiency (Xie et al. 2016), which is conducive to the improvement in green innovation efficiency (Qi and Liu 2023). However, too strong learning inertia can lead to enterprises relying too much on external resources to improve their own performance level (Li and Zeng 2019) while neglecting internal R&D technology and human capital investment (Kok et al. 2020). This shift in strategic focus could result in redundant digital resources that are difficult to absorb and utilized by the organization as soon as possible, thus triggering a series of problems such as the disconnection between the acquisition of external resources and the transformation of internal knowledge (Kok et al. 2020), which is not conducive to the green innovation of the enterprise (Tian et al. 2023). In addition, the staff's obsession with exploratory learning inevitably leads to inefficient application and a sharp increase in development costs (Li and Zeng 2019), which could reduce the team's confidence in acquiring environmental protection information and continuing R&D and innovation (Li and Zeng 2019). In summary, too strong learning inertia could weaken the "digital competence → green innovation performance" path to a certain extent. With regard to experience inertia, scholars at home and abroad have recognized its positive impact on innovation performance (Li and Zeng 2019; Liao et al. 2008; Cao et al. 2022; Yu et al. 2022), which is mainly via the empowerment of knowledge absorption (Cao et al. 2022), knowledge integration (Cao et al. 2022; Yu et al. 2022), and knowledge creation (Cao et al. 2022; Liu et al. 2017) to effectively promote enterprises to identify and predict market opportunities and "adaptive" business adjustment behavior, which in turn stimulates green innovation. When the inertia of experience exceeds a certain threshold, the enterprise's excessive adherence to the original procedures could limit the keen perception of market changes and cutting-edge digital technology (Xie et al. 2016). The "stable"

knowledge source could make the enterprise trapped in the shackles of outdated ideas, which could reduce the focus on energy saving and environmental protection technology innovation and ultimately fail to meet the market opportunities (Li and Zeng 2019; Tian et al. 2023). Based on the above, it is reasonable to speculate that the moderating effect of knowledge inertia may be different at different stages, and based on this, hypothesis H5 is proposed as follows.

**H5.** *Knowledge inertia plays an inverted U-shaped moderating role between digitization capability and green innovation performance.*

To sum up, the variable relationship structure model of this paper is shown in Figure 1.

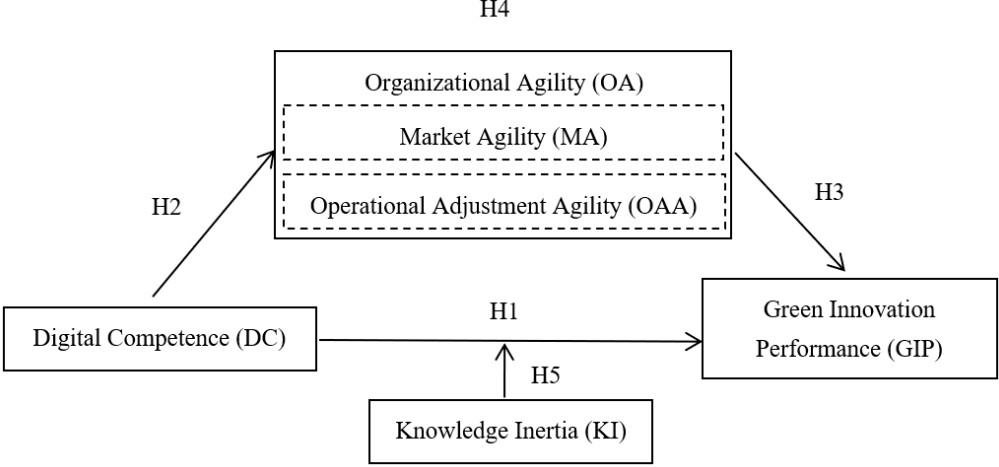

**Figure 1.** The research model.

## 4. Research Design

### 4.1. Sample Collection

The sample of the study is manufacturing enterprises implementing digital transformation and green innovation in China and given that digital transformation favors a macro approach under a strategic perspective, the study believes that on-the-job middle and senior managers in enterprises have more say. In the questionnaire design stage, we first searched for a mature scale that meets the purpose of the study based on the references; then, we optimized the semantics of the scale, the wording, and the order of the questions by combining the interview opinions of two experts who have long been engaged in the study of digital transformation, and three on-the-job middle and senior managers in manufacturing enterprises. For example, a lie detector question (please select "do not quite agree" for this question) was added to test the validity of the questionnaire. In addition, "Enterprises can carry out real-time dynamic analysis of services and resources for flexible adjustment" in the original maturity scale was revised to "enterprises can make dynamic adjustment according to real-time changes in services and resources" to ensure that subjects fully understand the meaning of the item. The pre-survey collected 50 questionnaires, and according to the feedback results, most of the subjects were unwilling to disclose the name of the enterprise, similar questionnaires can retain only one question, and there are ambiguities in individual questions, etc. The final questionnaire was revised again and passed the test. The final questionnaire was corrected again, 500 copies were distributed online via a professional research organization, and the questionnaires from different managers of the same enterprise were summarized and averaged before being put into the enterprise level for data analysis.

Excluding 117 invalid questionnaires, such as short answer time, obvious errors in answers, and answers from non-middle and senior managers, a total of 383 valid questionnaires were obtained, with an effective recovery rate of 76.6%. The statistical results show

that the sample enterprises are mainly distributed in Zhejiang (24.02%), Fujian (12.53%), Anhui (12.27%), and Liaoning (7.05%), which include both the relatively economically developed eastern coastal region and the slower developing central–western and northeastern regions, and the distribution range is somewhat representative and extensive. The sample concentrates on large- and medium-sized enterprises (90.34%) that have been established for five years or more (98.96%). In terms of the nature of ownership, joint ventures (58.49%) accounted for a relatively high proportion, followed by state-owned enterprises (30.55%). The surveyed enterprises are involved in seven categories of manufacturing industry segments, such as computer/electromechanical/electronic/mechanical and other high-end equipment manufacturing (31.85%), which possesses universality. With regard to personal background, there is a higher percentage of middle and senior managers with an education of a bachelor's degree or above (88.25%) and 5 years or more working in that company (73.1%), which indicates that the subjects have a better knowledge of the enterprise's digital capabilities and the company's development. This shows that the sample characteristics meet the requirements of this study.

*4.2. Variable Measurement*

This part will describe the scales used. The study adopted well-established domestic and international scales that have been used repeatedly and modified them to take into account the characteristics of the study. Except for the basic information of the enterprise, all the items of the scale are measured using a 7-point Likert scale, with scores from "1"–"7" gradually transitioning from "strongly disagree" to "strongly agree", and "4" representing neutral attitude "general".

Digital competence: Drawing on the maturity scale constructed by Yi et al. (2022) and improving it appropriately, it is measured in three dimensions: digital perception, operation, and resource collaboration. Each dimension has five question items.

Enterprise green innovation performance: mainly referring to the mature scale of Xing et al. (2020) and designing 8 measurement items after improving the expression appropriately.

Organizational agility: based on Lu and Ramamurthy's (2011) study, measured from two dimensions of market agility and operational adjustment agility, with two and four items, respectively.

Knowledge inertia: based on the research of Li and Zeng (2019) and Cao et al. (2022), measured from two dimensions of learning inertia and experience inertia, with 4 items in each dimension.

Control variables: Based on the variable selection and research ideas of established scholars, four variables, namely years of establishment, industry type, nature of ownership, and enterprise size, are selected as control variables. The number of years of establishment adopts four levels, namely less than 5 years, 5–9 years, 10–19 years, 20 years and above, coded 1–4 in turn; industry type includes seven categories of subdivided industries, such as electric power/energy/chemical/chemical fiber manufacturing industry, coded 1–7 in turn; with reference to the type of ownership of enterprises in China, it is divided into four types, namely, state-owned, joint-venture, private, and foreign-funded, coded 1–4 in turn; and with reference to the "Statistics released by the National Bureau of Statistics Division of Large, Small, Medium and Micro Enterprises" issued by the National Bureau of Statistics, the size of enterprises is measured from three categories of small (less than 300 persons), medium (300–999 persons), and large (more than 1,000 persons) and coded 1–3 in turn.

## 5. Empirical Analysis

### 5.1. Common Method Bias

The results of the questionnaire in this study were repeatedly filled out by the same subject, so Harman's one-factor method was used to verify the results (Zhou and Long 2004). SPSS 27.0 was used to conduct exploratory factor analysis on all the question items, and the number of extracted factors was set to 1. It was found that the eigenvalues of seven factors were greater than 1, and the variation explained by the first factor was 19.764%.

According to Podsakoff et al. (2003), the percentage was less than the critical criterion of 40%. So, there was no serious common method bias.

### 5.2. Reliability and Validity Analysis

The study used SPSS 27.0 to analyze the reliability and validity of five variables: market agility, operational adjustment agility, digital competence, corporate green innovation performance, and knowledge inertia. The results are shown in Table 1: the Cronbach's alpha coefficients of the variables are over 0.5, and the combined reliability (CR) is over 0.7. According to the standard interpretation results on the SPSSAU Official Website (2023) and other literature that uses this method (Tian et al. 2023; Pan and Tian 2017), these results indicate that the questionnaire is more reliable. In terms of validity, the selected scales are all mature scales developed by well-known researchers at home and abroad, and the contents of the foreign language scales have been continuously revised and improved by literature review, bilingual mutual translation, small-scale testing, etc., so they have good content validity. In addition, the structural validity and convergence validity are also determined based on the interpretation of the results on the SPSSAU Official Website (2023) and other literature (Tian et al. 2023; Pan and Tian 2017). Via confirmatory factor analysis (CFA), it can be found that the KMO value of each variable is greater than 0.6, indicating that there is a correlation between the variables in question, which meets the requirement of structural validity. In addition, the average common factor variance (AVE) extracted from each variable is greater than the critical value 0.5, which indicates that the convergence validity is good.

**Table 1.** Reliability and validity analysis (N = 383).

| Variables | Measurement Item | Factor Loading | $\alpha$ | KMO | AVE | CR |
|---|---|---|---|---|---|---|
| Digital capability DC (Yi et al. 2022) | Enterprises can see and identify data sources that have business value | 0.956 | 0.903 | 0.925 | 0.594 | 0.904 |
| | Enterprises can keep abreast of the latest information about external technology research and development or product production | 0.662 | | | | |
| | Enterprises can detect changes in the market competitive environment based on big data | 0.633 | | | | |
| | Enterprises can more accurately judge their own level of digitization | 0.692 | | | | |
| | Enterprises can match digital improvement schemes according to the strength of their management capabilities | 0.562 | | | | |
| | Companies can analyze digital information abstractly for precise market positioning | 0.67 | | | | |
| | Enterprises can use digital means to optimize business processes and resource allocation | 0.633 | | | | |
| | Enterprises can provide digital marketing management strategies for market analysis and customer experience | 0.741 | | | | |
| | Enterprises are able to dynamically adjust to real-time changes in services and resources | 0.574 | | | | |
| | Enterprises improve digital tools and components to improve the efficiency of business intelligence decisions | 0.58 | | | | |
| | The enterprise service systems have unified information exchange interfaces or modes | 0.653 | | | | |
| | Enterprises can aggregate internal and external digital resources according to innovation needs | 0.569 | | | | |
| | The enterprise can share internal and external information owned by the organization according to the need for cooperation | 0.747 | | | | |
| | Achieve good coupling interaction or diversified assistance between enterprises and stakeholders | 0.582 | | | | |
| | The enterprise can optimize the key processes of the organization | 0.583 | | | | |

**Table 1.** *Cont.*

| Variables | Measurement Item | Factor Loading | α | KMO | AVE | CR |
|---|---|---|---|---|---|---|
| Green innovation performance GI (Xing et al. 2020; Zameer et al. 2020) | In the past two years, companies have developed new products and services in environmental management | 0.65 | 0.521 | 0.665 | 0.527 | 0.753 |
| | Companies choose less polluting materials for product development or design | 0.687 | | | | |
| | Companies choose materials that consume the least amount of energy and resources for product development and design | 0.531 | | | | |
| | During the production process, companies carefully evaluate whether the product is easy to recycle, reuse, and decompose | 0.441 | | | | |
| | The production process of the enterprise effectively reduces the emission of harmful substances or waste | 0.517 | | | | |
| | Companies recycle waste and discharge it during production for disposal and use | 0.459 | | | | |
| | The production process of the enterprise consumes less water, electricity, coal, or oil | 0.695 | | | | |
| | The production process of the enterprise effectively reduces the use of raw materials | 0.524 | | | | |
| Organizational Agility (Lu and Ramamurthy 2011) | Market agility MA | | | | | |
| | Businesses are able to respond quickly and meet the special needs of customers | 0.885 | 0.772 | 0.615 | 0.639 | 0.778 |
| | Companies can rapidly expand or shrink production service levels in response to fluctuations in market demand | 0.71 | | | | |
| | Operational adjustment agility OAA | | | | | |
| | Companies can quickly make the necessary alternative arrangements and internal adjustments to cope with supply disruptions | 0.86 | 0.95 | 0.849 | 0.829 | 0.951 |
| | Businesses are able to quickly make and implement appropriate decisions in response to market/customer changes | 0.872 | | | | |
| | Companies are able to continuously transform or reorganize their organizations to better serve the market | 0.936 | | | | |
| | Businesses are able to see changes in the market and complexity of the environment as opportunities to invest quickly | 0.976 | | | | |
| Knowledge Inertia KI (Li and Zeng 2019; Cao et al. 2022) | Businesses are used to learning new concepts and new approaches | 0.537 | 0.834 | 0.84 | 0.586 | 0.849 |
| | Businesses are used to learning new things | 0.564 | | | | |
| | Companies are used to exploring the knowledge of external organizations | 0.615 | | | | |
| | Businesses are used to solving problems in different ways | 0.613 | | | | |
| | Businesses are used to getting knowledge from a fixed source | 0.786 | | | | |
| | Companies tend to rely on past experience | 0.834 | | | | |
| | An organization's past knowledge and experience can affect the acceptance of new knowledge | 0.731 | | | | |
| | Businesses are used to constantly leveraging existing knowledge | 0.763 | | | | |

*5.3. Correlation Analysis and Multicollinearity Test*

In this study, Pearson's correlation coefficient method was used to test the correlation, mean, standard deviation, and variance inflation factor between the variables. According to the collinearity diagnostic criteria of Hair et al. (1995), the variance inflation factors for both the control and independent variables were less than 5, indicating that there was no serious problem of multicollinearity between the variables. For cases where the same dimension included multiple question items, the average function was used to combine multiple scale questions. As shown in Table 2, the positive correlations between digital capability ($r = 0.135$, $p < 0.001$), market agility ($r = 0.131$, $p < 0.01$), and operational adjustment agility ($r = 0.114$, $p < 0.01$) and green innovation performance are all significant, and the positive correlations between digital capability and market agility ($r = 0.131$) and operational adjustment agility ($r = 0.108$) are all at the 0.01 level of significant positive correlation, and this finding preliminarily validates the related hypotheses.

**Table 2.** Table of phase relations and colinearity statistics (N = 383).

| | Year | Ownership | Industry | Size | DC | MA | OAA | KI | GIP |
|---|---|---|---|---|---|---|---|---|---|
| Year | 1 | | | | | | | | |
| Ownership | 0.077 | 1 | | | | | | | |
| Industry | 0.029 | 0.04 | 1 | | | | | | |
| Size | −0.027 | −0.23 *** | 0.068 | 1 | | | | | |
| DC | −0.009 | 0.069 | 0.075 | 0.01 | 1 | | | | |
| MA | −0.054 | −0.04 | 0.012 | 0.044 | 0.131 ** | 1 | | | |
| OAA | −0.076 | −0.022 | 0.005 | 0.036 | 0.108 ** | 0.958 *** | 1 | | |
| KI | 0.005 | 0.016 | −0.005 | −0.027 | −0.11 ** | −0.286 *** | −0.275 *** | 1 | |
| GIP | −0.038 | 0.07 | 0.044 | −0.024 | 0.135 *** | 0.131 ** | 0.114 ** | −0.206 *** | 1 |
| Mean | 2.7 | 1.85 | 3.56 | 2.23 | 5.687 | 5.78 | 5.518 | 5.436 | 5.823 |
| S.D. | 0.638 | 0.72 | 1.446 | 0.609 | 0.68 | 1.202 | 1.378 | 0.897 | 0.506 |

Note: ** means $0.001 < p < 0.01$, *** means $p < 0.001$.

*5.4. Hypothesis Testing*

The study used SPSS 27.0 to conduct hierarchical regression analysis to test the validity of the main effect of digital capability on firms' green innovation performance, the mediating effect of organizational agility (market agility and operational adjustment agility), and the moderating effect of knowledge inertia, and the results are shown in Table 3. Model 1 shows that none of the control variables are significant on the dependent variable, and after adding the independent variables, the explanatory power of Model 2 increases, and digitization capability shows a significant positive effect on corporate green innovation performance ($\beta = 0.095$, $p < 0.05$), and hypothesis H1 is valid. In addition, according to Model 3 and Model 5, it can be seen that both market agility ($\beta = 0.056$, $p < 0.05$) and operational adjustment agility ($\beta = 0.042$, $p < 0.05$) significantly and positively affect the green innovation performance of enterprises, and Hypothesis H3 are validated.

**Table 3.** Regression analysis result (N = 383).

| Variables | GIP | | | | | | | | MA | | OAA | |
|---|---|---|---|---|---|---|---|---|---|---|---|---|
| | M1 | M2 | M3 | M4 | M5 | M6 | M7 | M8 | M9 | M10 | M11 | M12 |
| Year | −0.036 | −0.034 | −0.03 | −0.03 | −0.029 | −0.028 | −0.012 *** | −0.038 | −0.095 | −0.091 | −0.16 | −0.157 |
| Ownership | 0.048 | 0.042 | 0.051 | 0.045 | 0.049 | 0.043 | 0.077 ** | 0.039 | −0.057 | −0.073 | −0.018 | −0.033 |
| Industry | 0.015 | 0.012 | 0.015 | 0.012 | 0.015 | 0.012 | −0.033 | 0.016 | 0.01 | 0.002 | 0.005 | −0.002 |
| Size | −0.01 | −0.013 | −0.014 | −0.016 | −0.013 | −0.015 | 0.043 | −0.013 | 0.067 | 0.061 | 0.07 | 0.065 |
| DC | | 0.095 * | | 0.084 ** | | 0.087 ** | 0.013 | 0.38 ** | | 0.236 ** | | 0.220 * |
| MA | | | 0.056 * | 0.05 ** | | | | | | | | |
| OAA | | | | | 0.042 * | 0.037 ** | | | | | | |
| KI² | | | | | | | −0.016 | 0.042 | | | | |
| DC × KI² | | | | | | | | −0.009 ** | | | | |
| R² | 0.219 | 0.325 | 0.326 | 0.439 | 0.322 | 0.335 | 0.373 | 0.284 | 0.206 | 0.324 | 0.237 | 0.429 |
| Adjusted R² | 0.191 | 0.212 | 0.213 | 0.323 | 0.209 | 0.218 | 0.258 | 0.196 | 0.105 | 0.211 | 0.193 | 0.314 |
| F | 2.83 | 3.931 | 2.035 | 2.514 | 3.659 | 2.271 | 4.94 | 4.882 | 3.562 | 4.815 | 2.663 | 3.43 |

Note: * means $0.01 < p < 0.05$, ** means $0.001 < p < 0.01$, *** means $p < 0.001$.

Referring to the steps of testing the mediating effect by Li et al. (2022) and Yi et al. (2022), the analyzing process of this study is as follows: (1) Regression of mediating variables on independent variables. The results of model 10 and model 12 show that digitalization capability has a significant positive effect on the mediating variable market agility ($\beta = 0.236$, $p < 0.01$) and also on operational adjustment agility ($\beta = 0.220$, $p < 0.05$), so the hypothesis H2 are tested; (2) Testing the regression of the dependent variable on the independent variable. According to the results of model 2, it can be seen that this condition holds; (3) The dependent variable is regressed on both the independent variable and the mediator variable, and the regression coefficients of the independent variable and the mediator variable are observed. Compared with model 2, the regression coefficient of the independent variable in model 4 is significant and reduced (from $\beta1 = 0.095$, $p < 0.05$ to $\beta2 = 0.084$, $p < 0.01$), and the regression coefficient of the mediator variable reaches

a significant level (β = 0.05, $p < 0.01$), so market agility plays a partially mediating role; similarly, the addition of operationally adjusted agility to model 6 reveals a significant and reduced regression coefficient (from β2 = 0.084, $p < 0.01$) for the independent variable. The regression coefficient is significant and decreasing (from β1 = 0.095, $p < 0.05$ to β2 = 0.087, $p < 0.01$), and the regression coefficient of the mediator variable reaches a significant level (β = 0.037, $p < 0.01$); therefore, operational adjustment agility also plays a partial mediating role. Therefore, hypothesis H4 is tested. In order to reduce the effect of data bias, the variables related to the moderating role are centered before regression analysis. Model 8 is constructed by combining the views of Fang et al. (2015): in the analysis box, the dependent variable, the control variable, the independent variable, the quadratic term of the moderating variable knowledge inertia, and the interaction term (digitalization ability × knowledge inertia square) are put in turn. According to Table 3, the quadratic term of knowledge inertia has a significant inverted U-shaped moderating effect (β = −0.009, $p < 0.01$) on the path of "digitization capability → enterprise green innovation performance", and the hypothesis H5 is valid.

*5.5. Robustness Test*

To ensure consistency and stability of the above explanations, according to the Bootstrap method used by Cao et al. (2022), this study utilized the PROCESS plug-in to further test the robustness of the main and mediating effects, choosing MODEL4 with a sample size of 5000 and a confidence level of 95%. The results, as shown in Table 4, show that all direct effect relationships were positive and significant, and none of the 95% confidence intervals contained 0. Thus, H1, H2, and H3 were again confirmed. Both mediating effect paths were significant, and none of the 95% confidence intervals contained 0. Both market agility (β = 0.012, $p < 0.05$) and operational adjustment agility (β = 0.008, $p < 0.05$) acted as partial mediators; thus, hypothesis H4 was reconfirmed. In addition, the study again tested the moderating effect via Model 1 in the Process plug-in and found that digitization capability was negatively significant for green innovation performance under the effect of the quadratic term of knowledge inertia. This suggests that within a certain range, knowledge inertia strengthens the main effect path, and above a certain threshold, it hinders that effect path, and hypothesis H5 is re-confirmed.

**Table 4.** Test result of Bootstrap method.

| Path | Effect | Efficiency Value | SE | 95% Bias-Corrected CI | |
| --- | --- | --- | --- | --- | --- |
| | | | | LLCI | ULCI |
| DC → GIP | Direct effect | 0.095 ** | 0.036 | 0.021 | 0.165 |
| DC → MA | Direct effect | 0.236 ** | 0.09 | 0.057 | 0.404 |
| DC → OA | Direct effect | 0.220 * | 0.107 | 0.009 | 0.424 |
| MA → GIP | Direct effect | 0.05 * | 0.023 | 0.014 | 0.101 |
| OA → GIP | Direct effect | 0.037 ** | 0.019 | 0.005 | 0.079 |
| DC → MA → GIP | Intermediary effect | 0.012 * | 0.022 | 0.002 | 0.031 |
| DC → OAA → GIP | Intermediary effect | 0.008 * | 0.019 | 0.001 | 0.024 |

Note: * means $0.01 < p < 0.05$, ** means $0.001 < p < 0.01$.

*5.6. Heterogeneity Test*

This part will further analyze the role of firm age, ownership type, segmentation type, and firm size on the dependent variables, comparing the mediating and moderating effects. Firms are classified into three types: low (0–9 years), medium (10–19 years), and high (20 years and above) according to firm age; sample firms are classified into state-owned and non-state-owned according to ownership type; and are divided into labor-intensive and technology-intensive industries according to the differences in the technological level of the manufacturing industry: the labor-intensive industries include the manufacturing industries of apparel, toys, and light foodstuffs, and the technology-intensive industries Including computer, electronics, chemical, aerospace, metal, medicine and health and other

manufacturing industries; company size will be divided into small enterprises with less than 300 people, more than 300 people are divided into large- and medium-sized.

The mediated effect values of market agility are 0.081, 0.036, and 0.246 for low, medium, and high ages, respectively, and the mediated effect values of operational adjustment agility are 0.057, 0.03, and 0.208, respectively, indicating that firms established for 20 years and above pay more attention to improving green innovation performance via organizational agility; the mediated effect values of market agility and operational adjustment agility for state-owned firms are 0.057 and 0.043 for state-owned enterprises and 0.074 and 0.054 for non-state-owned enterprises, respectively, because non-state-owned enterprises are more resilient to adapt to the competitive market economy and rely more on organizational agility to enhance green innovation performance; the mediation effect values of market agility and operational adjustment agility for large- and medium-sized enterprises are 0.061 and 0.045, respectively, and the mediation effect is not significant for small enterprises The mediating effect of market agility and operational adjustment agility in labor-intensive manufacturing industries is 0.026 and 0.014, respectively, while that in technology-intensive industries is 0.008 and 0.007, respectively, indicating that industries with greater personnel mobility need to improve organizational agility to achieve green innovation. The moderating effects of knowledge inertia on low, medium and high enterprises of three different ages are −0.012, −0.003, and −0.064, respectively, indicating that high age enterprises should pay more attention to the moderating effect of knowledge inertia; the moderating effects of state-owned enterprises and non-state-owned enterprises are −0.009 and −0.01, respectively, because state-owned enterprises are relatively less flexible and have a more complex technological structure, and the employee's The moderating effect on the performance of state-owned enterprises and non-state-owned enterprises is −0.009 and −0.01, respectively, because state-owned enterprises are relatively less flexible and have more complex technological structures, and the impact of employees' learning initiative on performance is smaller than that of non-state-owned enterprises, while the impact of employees' constraints imposed by past experience and habits is larger than that of non-state-owned enterprises beyond a certain threshold. The moderating effect on small enterprises is −0.003, and that on medium and large enterprises is −0.012, which means that the larger the enterprise is, the more likely it is to be affected by knowledge inertia.

## 6. Conclusions and Insights

The study finds that, firstly, enterprises with higher digitization capabilities promote green innovation performance and work via the agility of their response to the external market and the adjustment of their internal operation processes. The level of digitization empowers green production, transforms traditional and inefficient business management models, reshapes the enterprise value system, upgrades the marketing service platform via the automation and intelligence of product production and manufacturing, and encourages enterprises to develop a sound mechanism for sustainable innovation; at the same time, the enterprise's green innovation performance indexes are closely related to the operation process, social responsibility image, and value system, which in turn empowers digitization capabilities to make improvements. The ability of enterprises to perceive risks, respond to changes such as demand fluctuations, and make timely adjustments in a dynamic environment is an important way to improve green innovation performance in the era of the digital economy. In practice, the accumulation of digital knowledge and the improvement in technological level help enterprises to perceive business opportunities, establish an environmental image, and stimulate enterprises to realize cost reduction and efficiency, which in turn empowers enterprises to create greater environmental value to a certain extent, and stimulates the internal green innovation elements to burst out, thus injecting new vitality into green innovation. Secondly, there is an inflection point in the regulation mechanism of knowledge inertia on the main effect, indicating that moderate knowledge inertia can stimulate enterprises to empower digitization to improve the performance of green innovation, and once it exceeds a certain threshold, it could increase the cost of the

enterprise, reduce the operational efficiency, and weaken the path of this effect. This means that in order to adapt to the ever-changing competitive environment, enterprises should make full use of the advantage of knowledge inertia to stimulate the subjective learning initiative of employees so that the speed of green innovation matches the development of digitization and, at the same time, be highly alert to the stereotyping effect that knowledge inertia may bring to innovation, such as stereotyping and conforming to rules. Third, enterprises that have been established for 20 years or more, non-state-owned enterprises, labor-intensive enterprises, and large and medium-sized enterprises rely more on organizational agility to enhance green innovation and should pay more attention to the moderating role of knowledge inertia.

The theoretical contributions of the study include the following three points: Firstly, scholars generally believe that digital capability could effectively improve innovation performance (Shen et al. 2021), and this promotion effect is reflected in reducing opportunity cost (Li et al. 2022), promoting human capital upgrading (Zhang and Du 2022), improving corporate governance (Xiao et al. 2022), and increasing investment in corporate innovation (Michael et al. 2020). However, very few scholars, such as Luo et al. (2022), have discussed the impact of enterprise digitalization on green innovation; this paper forms a certain supplement. Secondly, by exploring the mediating role played by organizational agility, the paper provides new ideas for a deeper understanding of the specific path of digital technology on firms' innovation capabilities. It not only enriches the research on antecedent variables of organizational agility by proving that the level of digitalization promotes market agility and operational adjustment agility but also overcomes the shortcomings of more studies that view organizational agility as a single dimension. Thirdly, the inverted U-shaped influence mechanism of knowledge inertia and green innovation performance is enriched. Optimizing the performance of enterprise green innovation has always been the core research direction of the academic circle for the long-term exploration of enterprise reform and development. Most of the existing studies explored the green innovation performance from the organizational level, such as senior executives' environmental awareness (Yuan and Li 2023) and green human resource management (Yu et al. 2022), or from the macro perspective, such as government subsidies (Wang 2023) and green tax reform (Yu et al. 2023). Some scholars have also discussed the negative regulatory effect of organizational inertia on the green transformation of enterprises (Pan and Wang 2022), while there is not enough research starting from the factors that have a non-unidirectional relationship with green innovation performance, such as knowledge inertia. In the existing theoretical and extended research of knowledge, the phenomenon of knowledge inertia in manufacturing enterprises has not aroused more attention and research enthusiasm. The formation and evolution of its related theoretical system need to be studied continuously. Therefore, this paper analyzes the regulatory mechanism of knowledge inertia on "digital capability-corporate green innovation performance", which has certain theoretical significance.

The study also has some practical guidance: first, governments should increase their efforts to encourage enterprises to utilize the latest information technologies, such as artificial intelligence, big data, and cloud computing, in their business processes and to support initiatives such as change consulting, financial investment, piloting, and service improvement, in order to improve the efficiency of business operations and green innovation performance. Secondly, enterprises should strengthen their skills in dealing with uncertainty and reinforce the concept of synergistic development of technology and business. Enterprises can quickly retrieve effective information from various modern information dissemination tools such as official websites, micro-blogs, and WeChat platforms to better understand customer needs. Furthermore, knowledge inertia, as an intrinsic attribute of enterprises, affects all aspects of growth, and the moderate control of knowledge inertia can effectively improve the green innovation performance of enterprises. In addition, for long-established, labor-intensive, and large-scale enterprises, it is more important to pay attention to the

dual impact of knowledge inertia, which should be adjusted to an appropriate position according to the enterprise's own situation.

However, there are some shortcomings in this study. First, the study does not discuss based on other theoretical perspectives such as knowledge management theory and organizational behavior theory; second, the mediating and moderating variables are also relatively single, and it is necessary to continue to explore the role of variables such as corporate culture, data security, organizational structure, etc.; furthermore, the study obtains first-hand data in the form of questionnaires based on the perceptions of middle and senior managers on the characteristics of the level of digital development, green innovation, and knowledge behavior argumentation, there is a certain measurement error, and it is possible to design A and B papers combined with the responses of first-line managers and grassroots employees to reverse the validation, which will be supplemented and analyzed in the subsequent study.

**Author Contributions:** Z.J.—Conceptualization, Funding acquisition, Methodology, Project administration, Software, Supervision, and Writing—review and editing; Y.Z.—Data curation, Formal analysis, Investigation, and Writing—original draft; H.G.—Resources, Validation, and Visualization. All authors have read and agreed to the published version of the manuscript.

**Funding:** This research was funded by the Fundamental Research Funds for the Central Universities [2023JBW8002].

**Institutional Review Board Statement:** Not applicable.

**Informed Consent Statement:** Informed consent was obtained from all subjects involved in the study.

**Data Availability Statement:** Due to privacy concerns, the data used in this study will not be uploaded. For more data supporting records, please contact the author at 23120735@bjtu.edu.cn and indicate intent.

**Conflicts of Interest:** The authors declare no conflict of interest.

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
