# Peer review of "A Study of the Impact of Digital Competence and Organizational Agility on Green Innovation Performance of Manufacturing Firms—The Moderating Effect Based on Knowledge Inertia"

_admsci, doi:10.3390/admsci13120250_

Round 1
Reviewer 1 Report
Comments and Suggestions for Authors
Thank you for the opportunity to read this very interesting article. The impact of digital competence and organisational agility on the success of green innovation is a very important topic. In order to provide new findings, the researchers collected a large sample of manufacturing companies and used hierarchical regression. The results are informative, but it is not clear what contribution the paper makes. My main concerns are as follows:
· Although the authors have to some extent presented a research gap in the introduction, it is not clear what the aim of the study is. I therefore suggest that at least the research questions should be added. In addition, several passages are not supported by relevant literature (lines 39, 48, ...).
· The theoretical basis is very brief and could be included in the section Conceptual basis and hypotheses. Regarding the research model, why was only the moderating role of KI considered between DC and GIP? What about the moderating role between DC and OA and OA and GIP?
· The research design is appropriate. The companies were selected purposively, which is common for studies that focus in depth on a relatively small sample, which is not the case here. This technique is prone to research bias, so why was it chosen? Has this limitation been taken into account when generalising the findings?
· There is no explanation as to why hierarchical regression analysis is used in this study. In cases where the data sample is quite large, it is usually more appropriate to use SEM. Please explain.
· The discussion is very weak, with only limited reference to the existing literature and very limited discussion of the contribution of the study, especially the theoretical contribution.
Good luck with further research.
Comments on the Quality of English Language/
Reviewer 2 Report
Comments and Suggestions for Authors
Thank you for the opportunity to review your manuscript. The article takes up an important issue and has good potential. However, I have a number of concerns regarding your paper, that should be addressed in order to improve the contributions of your work. It is important to note that my concerns are more about the form than the content.
I hope you find my comments useful for the development of your research.
In general
- "Sometimes, the author(s) make categorical statements without theoretical support. This can be rectified by using more parsimonious language, as is expected in scientific texts. For instance, in the excerpt from lines 244-245, 'which will reduce the team's confidence [...]', the expression 'which will reduce' should be replaced with 'which could reduce'. Check for this kind of improvement throughout the entire text."
- "Sometimes, the author(s) use multiple sources within a single citation to support a sentence with diverse information. This raises the question: Did all the authors convey the same ideas? I believe they did not. Combining multiple works within a single citation can make it impossible to discern who expressed which particular information. To address this issue, we have highlighted some of these concerns in comments 3.3, 3.4, and 3.5 below."
Abstract
- The sentence “Based on the data from a large sample […] theory is applied to SPSS 27.0.” should be after the research aim “Hierarchical regression was used to empirically investigate the impact of digital capabilities […] effect of 9 knowledge inertia” to enhance the clarity and to provide a more comprehensive understanding for the reader.
- I think that “The conclusions not only deepen the academic theoretical research on the dual impact of knowledge inertia” can be replaced by the conclusion itself.
- Initiates the phrase “but also provide new ideas for companies to develop organizational […] advantage position” with “this paper contributes…”
1. Introduction
1.1. Lack of specific citations to support many claims in the introduction.
- Lines 27-29: “Innovation is the inexhaustible driving force […] while realizing economic development.”
- Lines 29-33: “The innovation of green economy is […] of digitization and greening.”
1.2. Lines: 62-63 “, which can provide some reference basis for academic research.” It is not necessary.
2. Theoretical basis
2.1. Lack of specific citations to support many claims in the section.
- 79-80 “Dynamic capability theory emphasizes the […] with changing market demands.”
- 80-82 “Dynamic refers to the need […] ever-changing external environment”
- If the above-mentioned excerpts were based on Teece et al (1997), please cite them adequately. Information by information needs to have its citation tracked. There can be no doubt about which information is based on which references may exist.
3. Conceptual background and hypotheses
3.1. Lack of specific citations to support many claims in the section.
- 101-102: “and the reduction of the pressure of financing […] to develop green innovations.”
- 103-104: “The reduction of financing pressure makes […] innovation and development.”
- 128-132: “For example, every time a consumer browses on the Taobao platform […] for the company”
- 136-139: “Business operations based on outdated […] thus improving enterprise agility”
- 152-156: “In a dynamically changing competitive market […] risk of green innovation, and improve the performance level.”
- 165-169: “At the same time, real-time business adjustments drive […] friendly and healthier raw materials, etc.”
- 194-195: “Higher market agility implies that firms are able to significantly improve their […] flexibility to respond to changes.”
- 195-199: “In fact, after successfully acquiring real-time market information with the help of data […] to achieve green innovation.”
- 199-202: “On the other hand, Operational Adjustment Agility focuses on an organization's ability […] which in turn affects team performance.”
- 230-234: “Driven by the willingness to learn, the enterprise focuses […] which is conducive to the improvement of green innovation efficiency”
- 237-242: “This shift in strategic focus will result in redundant digital […] which is not conducive to the green innovation of the enterprise.”
- 244-245: “which will reduce the team's confidence […] continuing R&D and innovation.”
- 253-257: “When the inertia of experience […] ultimately fail to meet the market opportunities.”
3.2 In line 202-207 “Firms with higher digital capabilities tend to have […] allocation upstream and downstream of the supply chain (Zhang et al., 2023;Zhang et al., 2021;Qi&Liu, 2023; Song et al., 2022)” have so many citations, but it is important to clarify who say what. Did all the authors say the same thing?
3.3 Once again, it is important to clarify who said what in the following excerpt from lines 221-225: “Learning inertia emphasizes the process in which employees are driven by the willingness to learn and acquire new knowledge […] procedural patterns and customary regulations (Li&Zeng, 2019; Liao et al., 2008; Cao et al.,2022)”. Did all the authors say the same thing?
3.4. The same commentary 3.3 applies to the excerpt from lines 225-230: “When employees' learning inertia changes from weak to strong, their willingness to explore, acquire and […] on innovation performance (Li&Zeng, 2019; Yi&Cao, 2022; Akter et al., 2016)”
3.5. The Figure 1 is not cited in the text. All figures and tables should be cited in the main text as Figure 1, Table 1, etc.
4. Research Design
4.1 Check the sentence from the lines 289-295 “[…] the original maturity scale was amended from […]the meaning of the question.”
4.2. fix some of the typing error in line 325 “by likert7 scale”
4.3. Is the excerpt from lines 326-328 repeated? “A score of "1"-"7" gradually transitions from […] attitude of "general".
4.4. Enter text between lines 328-329 stating that it will describe the scales used. For example, “below, we describe the scales used”. This will improve reading and comprehension of the paper.
4.5. Missing details or explanations about the data analysis, making it difficult for readers to follow the research methodology. This is fundamental to setting the parameters for data analysis and its references. For example, what support the statement “there was no serious common method bias” in line 360-361 taking into account the “critical criterion of 40%”?
5. Empirical analysis
5.1. Lack of specific citations to support many claims along the section:
- 360-361 “which was less than the critical criterion of 40%, so there was no serious common method bias”.
- 365-367 “The Cronbach's alpha coefficients of the variables are over 0.5, and the combined reliability (CR) is over 0.7, indicating that the questionnaire is more reliable
- 371-374 “through the validated factor analysis (CFA), it can be found that the KMO value of each variable is greater than 0.6, “[…] of structural validity’
- 374-375 “In addition, the average common factor variance extracted (AVE) […] critical value of 0.5, indicating good convergent validity”
- 380-382 “The variance inflation factors for both the control and independent variables were less than 5, indicating that there was no serious problem of multicollinearity between the variables”
Round 2
Reviewer 1 Report
Comments and Suggestions for Authors
Thank you for the updated paper and additional clarifications. My main concerns have been taken into account.
Author Response
Thank you for your timely feedback and your meaningful comments again.
With best wishes!